# The Effects of the COVID-19 Pandemic in Oncology Patient Management

**DOI:** 10.3390/ijerph19159041

**Published:** 2022-07-25

**Authors:** Mario Forrester, Luiza Breitenfeld, Miguel Castelo-Branco, Jorge Aperta

**Affiliations:** 1Faculty of Health Sciences, Universidade Da Beira Interior, Av. Infante D. Henrique, 6200-506 Covilhã, Portugal; luiza@fcsaude.ubi.pt (L.B.); mcbranco@ubi.pt (M.C.-B.); 2Sousa Martins Hospital, Avenida Rainha Dona Amélia, 6300-858 Guarda, Portugal; aperta.jorge@gmail.com

**Keywords:** value-based healthcare, oncology, patient management, COVID-19

## Abstract

The COVID-19 pandemic has changed the way cancer patients should be managed. Using published literature on best practices on oncology patient management, we developed checklists to establish which recommendations were followed and differences between healthcare staff and institutions in a local health unit (overseeing two regional hospitals and 14 primary Healthcare Centers) in an interior region in Portugal. Checklists were delivered and completed by 15 physicians, 18 nurses and 5 pharmacists working at the Hospitals, and 29 physicians and 46 nurses from primary healthcare centers. Hospital staff do not show statistically significant differences regarding most proposed recommendations for the oncology clinical pathway, human resources, treatments, patient management and service management. Primary healthcare centers seem to follow a similar trend. As a local health unit, general recommendations for Oncology Patient Management show statistically significantly different values on education of suspected cases, identification, isolation procedures and samples collection; extension of work schedules; and education on cancer patient and COVID-19 positive referral procedures. All the checklists indicated good-to-high internal consistency. Our analysis showed cohesive work between groups regarding control and prevention of sources of infection; therefore, it is considered the highest priority to ensure that all other services, including oncology, continue functioning. Patient management measures such as adjustments in treatments, analysis, patient care, referrals and emergencies were not ranked higher by responders.

## 1. Introduction

The COVID-19 pandemic context has deeply changed patient management, especially at-risk populations such as oncological patients. The COVID-19 pandemic has significantly overloaded hospital and healthcare systems throughout many countries. Material and human resources have been reorganized to manage the influx of patients requiring healthcare. Consequently, the pandemic led to an abrupt change in routine medical care of chronic and vulnerable populations such as cancer patients, whose outcomes depend on timely, relevant, and high-quality multidisciplinary interventions. In 2020, around 6 million COVID-19 cases have been confirmed worldwide and over 370,000 people perished because of it [1,2,3].

Cancer itself can affect the immune system by spreading into the bone marrow, making cancer patients more susceptible to COVID-19 infections. Moreover, cancer treatments also suppress other rapidly growing cells such as white blood cells, including lymphocytes T and C in bone marrow [4,5].

Several studies show that cancer patients suffer from increased complications and overall risk of death. Compared to the general population, these patients have a three-fold vulnerability to death due to COVID-19. The probabilities of needing to use invasive mechanical ventilation and observing high-risk critical symptoms increase when compared to non-cancer patients. As the pandemic advances into its third year and new variants arise, choosing an optimal vaccination strategy to protect cancer patients remains a priority. However, a recent study observed lower responsiveness to two-dose mRNA vaccinations in the cancer population, suggesting the use of booster vaccines to overcome this issue [5,6]. 

However, other studies state that even though patients with malignancy and active anti-tumor treatment are considered at high risk, the data in this context are conflicting and there is evidence clearly demonstrating no significant effect of anticancer therapies on COVID-19 mortality and morbidity, with some of these studies overlooking the effect of age on the latter and others stating that 20% of these patients may suffer from an asymptomatic disease only evident through serological assessment. Moreover, several investigators have evaluated the effects of cancer on the natural history and prognosis of COVID-19. Their results may not be generalizable to all populations with different cancer epidemiology and practice; also, there is a lack of information for direct comparison of patients with cancer in terms of clinical manifestations and outcomes of COVID-19 since the impact of the virus is not limited to this population [7,8,9].

To understand the impact of COVID-19 and create proper clinical guidelines to face the challenges of this pandemic, the oncology community initiated a combined effort on collection and sharing of data at an exceptional rate. Thus, the number of publications about COVID-19 grew exponentially and, as of October 2020, approximately 60,000 publications on COVID-19 and cancer have been listed on PubMed. Various oncology societies and national authorities recommended a series of implementations for everyday practice on cancer care and OPM during the COVID-19 pandemic. Therefore, this paper intends to analyze the first step of any new guideline implementation: are the healthcare professionals aware of the new guidelines? To answer this question we developed a checklist, using public recommendations, focused specifically on healthcare staff directly involved with these patients [4,10]. 

## 2. Materials and Methods

To achieve our main goal, we designed a prospective study that intends to collect quantitative and qualitative data through a checklist focused on healthcare staff directly involved with oncological patients in a local health unit in an interior region in Portugal. This health unit consists of two regional hospitals (RHs) and 14 Primary Healthcare Centers (PHCs). This local health unit oversees a total area of 5518 km^2^, a total population of 164 212 habitants and a population density of approximately 29.5 habitants per km^2^. The local health unit ethics committee approved the application of the checklists in a healthcare professional (nurses, physicians and pharmacists) sample. The period of study ran from October 2020 until June 2021. 

From March 2020, COVID-19 deaths accounted for 5.8% of all deaths in Portugal, making it the second-biggest cause of death during that year. These rates were higher in the Northern regions (91.5 per 100 000 individuals) and in the Lisbon Metropolitan areas (71.5). The center region, which was where our study was conducted, observed a death rate of 57.8 per 100 000 in 2020. There are no reports of COVID-19 death rates for oncological patients in the national reports. Moreover, when our study began in October 2020, cases were rising considerably in all regions, especially during the winter months of December, January, and February. In the final months of our study, cases decreased in all regions [11,12].

The pandemic context forced an increase of 6.6%in government healthcare expenses. This re-organization, translated into an increase of 6.8% in healthcare staff costs (hirings, extra hours, among other) and 16% increase in costs of medium consumption (personal protective equipment and medication, among others). Furthermore, in 2020 the number of physicians per 100,000 habitants increased by 0.2% (5,6) compared to 2019 and the number of nurses kept the tendency to increase of 2.9% from past years. Despite the latter, the number of consultations (general and emergency), surgeries and hospitalizations decreased in 2020 for all major medical specialties except for infectious diseases, which registered more than 10,000 for 2020, more than double the amount from 2019 [13,14].

Based on a national validated database (PORDATA), we obtained the total number of physicians and nurses working at the local healthcare unit. The last update in 2019 revealed a total of 223 physicians and 555 nurses. A total of 15 physicians and 18 nurses from different departments directly involved with Oncology patients from the RH returned the questionnaire (N = 49, response rate = 67%); likewise, from the 10 pharmacists at the RH, 5 pharmacists involved with the hospital oncology service answered the questionnaire (N = 5, response rate = 100%). From the PHCs, a total of 29 physicians and 46 nurses returned the survey (Approximate N = 389, response rate= 19.2%). There were no pharmacists at the PHCs (N = 0). Only one of the two RHs had an oncology service; therefore, most of our hospital responders worked there.

The checklists contain several items to be answered by responders about which measures were developed or not developed at their work services (binary checklists). Since surveys were developed depending on the healthcare platform, the checklists were also adapted likewise, therefore we assessed the arrangement of the items with the help of other healthcare professionals and academics in open discussion groups to best categorize and validate the checklists. The first two checklists contained physical and technical–scientific parameters that apply mostly at hospital settings, the third one applies for both primary healthcare centers and hospitals (horizontal analysis) and the last one applies for pharmaceutical services. The development of the 4 checklist items was based on the literature available at the time and recommendations made by international and national oncology societies and groups [15,16,17,18,19,20,21,22,23,24,25,26,27,28,29,30,31,32,33,34,35].

The checklists subjects are shown as per the following:

**Physical Changes** in the OPM During the COVID-19 Pandemic (for RHs): 16 items focused on modifications of the actual clinical pathway and human resources.

**Technical-Scientific Changes** in the OPM During the COVID-19 Pandemic (for RHs): 28 items focused on modifications based on recommendations made by societies and groups on treatments, patient management and service management.

**General Changes** in Oncology Clinical Activities During the COVID-19 Pandemic (for RHs and PHCs): 17 items focused on general recommendations applicable to healthcare staff at hospitals and primary healthcare centers.

**Pharmaceutical Changes** in the Oncology Processes During the COVID-19 Pandemic (RH Pharmaceutical Services): 14 items focused recommendations on drug management, dispensing, human resources and access to medication.

A Mann–Whitney U test was selected to determine if there were differences between nurses and physicians from each healthcare platform in their day-to-day work during the COVID-19 pandemic, as well as differences in the workflow at the RH and their PHCs. Regarding the pharmacist checklist, we were not able to perform a Mann–Whitney U test since the checklist is different from other groups studied. Finally, Cronbach’s alpha was used to measure the internal consistency of the items from all checklists, except the pharmacist checklist due to a reduced sample (N = 5). By using this checklist approach, we hoped to obtain a more subjective measure of the level of agreement our responders had towards OPM during the pandemic.

### 2.1. Inclusion and Exclusion Criteria

Only permanent healthcare staff working at the local health unit were included in the study. Responders needed to answer the totality of the checklists and the questionnaire in order to participate. Hospital responders must work in coordination with the Oncology Department; therefore, the hospital services included in this analysis were Internal Medicine, Gastroenterology, Pneumology, Urology, Surgery and Palliative Care. All Primary Healthcare centers responders willing to participate were included in the study. New healthcare staff hirings during the COVID-19 pandemic were excluded from the study as they were not able to provide a point of comparison for OPM.

### 2.2. Patient and Public Involvement

This study is mainly focused on healthcare professional appreciations; hence, no patients were included. Participants were recruited by previous analysis of key deciders or individuals that were constantly working with cancer patients, making their input relevant to this investigation. Every participant was either informed personally or via an online introduction prior to starting the questionnaire. Questionnaires were delivered online and physically to collect the information. All answers were anonymous to create a safe and private environment for responders to answer freely in the open critic spaces at the end of the questionnaire. The results will be made public to all participants once published or on demand.

## 3. Results

To best explain and present our results, we have outlined several Mann–Whitney U tests to understand if there were statistical differences in the OPM between nurses and physicians at the local health unit. Thus, Table 1 shows a set of items for physical changes or recommendations proposed in the literature for the COVID-19 pandemic; the items are organized by combined response rate (nurses and physicians) for a developed recommendation. Additionally, Table 1 includes responders median answer distribution or median rank, the Mann–Whitney U-value and the *p* value. The lower the mean rank for one of the compared groups, the more responders answered positively for a developed recommendation (higher response rate).

The RH healthcare staff do not show statistically significant differences regarding their answers on the majority of proposed recommendations for the oncology clinical pathway and human resource. The only statistically significant value was regarding working in “mirror” teams with a *p* value of 0.033. Regarding the Cronbach’s alpha, our analysis indicates a value of 0.723.

Table 2 comprises the technical-scientific changes or recommendations proposed in the literature for OPM at hospitals during the COVID-19 pandemic; the items are also organized by combined response rates (nurses and physicians) for a developed recommendation with responders’ median answer distribution or median rank, the Mann–Whitney U-value and the *p* value. Responders do not show statistically significant differences regarding their answers on most proposed recommendations for treatments, patient management and service management. The only statistically significant value was regarding delaying visits and surveillance exams in patients considered non-urgent, with a *p* value of 0.018. The Cronbach’s alpha of this set of items indicates a value of 0.833.

In regard to the General Changes or recommendations for Oncology Clinical Activities During the COVID-19 Pandemic, Table 3 contains items applicable to PHC staff and responders’ median answer distribution or median rank, the Mann–Whitney U-value and the *p* value. Items are organized by combined response rates for a developed recommendation. Responders from the local health unit do not show statistically significant differences regarding their answers on most of the proposed recommendations. The only statistically significantly different value was regarding the decrease the number of routine visits with a *p* value of 0.012. The Cronbach’s alpha of this set of items indicates a value of 0.855.

As a local health unit (both the RH and PHCs), it is possible to determine which general recommendations were mostly developed and if there were differences between healthcare platforms. Table 4 includes the same items and analysis as Table 3, allowing a horizontal analysis. Responding healthcare staff from our local health unit do not show statistically significant differences regarding their answers on most proposed recommendations. The only statistically significantly different values were education on suspected cases, identification, isolation procedures and samples collection; extension of work schedules; and education on cancer patient and COVID-19 positive referral procedures; with *p* values of 0.027, 0.30, and 0.025, respectively. The Cronbach’s alpha of this set of items indicates a value of 0.828.

## 4. Discussion

This study intends to determine which of the international and national recommendations (physical, technical-scientific and general) regarding OPM during the COVID-19 pandemic were applied by healthcare staff (nurses and physicians) at our local health center. Regarding physical adjustments in the OPM at the RH, Table 1 shows that most responders worked in a very similar manner regardless of their department and their profession, implying a coordinated effort to best manage their cancer patients. However, not all the recommendations given were equally applied at the RH as per their response rates.

It seems that the priority for health care staff at the RH, regarding the actual oncology clinical pathway, was reducing possible ways of infection by decreasing visits, implementing non-presential consults, reorganizing human resources and patient management inside the hospital. Based on the response rates, modifications on analysis, patient treatment management and referrals were not completely modified and continue to function in a similar way throughout the pandemic. Extending work schedules, separating early detection areas, and designing new options for clinical pathways were not a priority to our responders. Finally, one physician explicitly mentioned on the “Other” option that they “test patients before each treatment”

Recommendations from Table 1 with some level of difference between our responders were in items 2, 5 and 7, according to the mean answer distribution. All the latter had higher response rates according to the nursing staff, which could be explained by greater number of nurses vs. physicians (regarding working in mirror teams), their tasks (measuring patients’ temperature before receiving treatment) and the resources they need to properly work since physicians might need more equipment for diagnoses and analyses.

Technical–scientific recommendations, as shown in Table 2, indicate a cohesive work between our analyzed groups. As well as Table 1, not all recommendations were applied with the same priority level as per their combined response rates. Likewise, Table 2 indicates that the priority for the RH healthcare staff during the COVID-19 pandemic relied, as discussed in Table 1, in containing all possible ways of infection through strict measures of control and prevention. Consequently, remote consults and follow-up, use and training in PPE, new triage protocols for COVID-19, delay non-urgent visits, educating staff on identifying suspected cases and assessing symptoms considering other causes of cancer seem to be more relevant to RH staff.

Furthermore, our responders considered as lower priority (nonetheless relevant) technical–scientifical recommendations on cancer and COVID-19 patient management regarding the following: adapt and communicate risk assessments, test patients for analysis and urgent surgery, develop new referral procedures, monitor symptoms through strict surveillance (especially for high-risk cases), patient management according to risk and treatment, treat oncological emergencies when the prognosis is not affected, and prefer oral medication schemes when possible. Finally, all other recommendation for treatments (chemotherapy, supportive, hormone therapy, growth factors, at home use, etc.), access to surgery, and access to clinical trials were not marked as higher priority.

Although our analysis showed only one recommendation with a statistically significant difference for Table 2, there were other items where the response rates implicate some level of non-cohesive work. As per the mean answer rank, more nursing staff had higher response rates for items 6, 7, 8, 10, 17 and 23, while more physicians seem keener for items 5 and 9.

This could be explained due to their specific functions in the team, as more nurses are involved initially with patients before they are assisted by physicians to, further on, be discharged by the nursing staff with all the details for further healthcare which could demand more attention and organization from the nursing staff (possible explanations for items 6, 7, 8, 10, 17 and 23). Regarding items 5 and 9, physicians are mostly responsible for coordinating visits, analyses and/or surgeries ensuring infection control protocols. Finally, all other items from Table 2 had a similar mean answer distribution.

General recommendations applied by PHC staff concerning oncology clinical procedures, as indicated in Table 3, show organized work between nurses and physicians regarding OPM. Similarly to the RH responders, PHC staff considered containing all possible ways of infection as high priority. Therefore, applying strict measures of infection control such as PPE use and training on its use, triage protocols for COVID-19, proper training on COVID case identification and their management, decreased routine visits, and delayed surveillance in non-urgent cases became common practices throughout the pandemic.

Other measures that were not higher ranked according to our responders were: adapt and communicate risk assessments, remote consult systems, cancer patient management according to risk and treatment, adequate access to resources, extension of work schedules, work in “mirror teams”, segregating early detection areas, monitorization of active cancer patients, multidisciplinary video-consultations, cancer and COVID-19 positive patient management, and create new clinical pathways designs. Finally, one nurse implied on the “Other” option that “There are no protocols each doctor does in his own way”.

Since PHCs at this local health unit do not have an oncology day hospital, their role with cancer patients relies on referral procedures whenever patients show up with a tumor suspicion, with physical complications from their treatment and/or treating general symptomatology; however, the COVID-19 pandemic forced them to readjust their processes for OPM.

Table 3 also shows some items where the mean answer distribution varied between groups (even though not all were statistically significant) indicating different roles or tasks within healthcare centers or even non-cohesive work. Regarding items 3, 5, 6 and 8, physicians had higher response rates, which could be due to their specific tasks since they are more involved (compared to nurses) in the triage protocols, patient visits, surveillance exams and consults. However, item 11 shows nursing staff with greater extension of their work schedules, which could be explained by a lack of professionals, a problem that, in the latest years, has become an issue in the rural areas of Portugal.

As an entire Health Unit, Table 4 measures the same application of items as Table 3, therefore our interest here is to determine the level of cohesive work between healthcare platforms using the same type of responders. Both PHC and RH staff agreed on the priority of infection control as the first six items from both tables have the same order of appearance. All the other items not ranked as higher priority seem to follow the similar trend in both tables with one nurse responder from a PHC indicating that “Suspected patients are transferred”.

However, there are recommendations that were not followed equally in both healthcare platforms, thus creating statistically significantly differences between groups. Table 4 indicates that information on suspected cases logistics and extension of work schedules were significantly more applied at PHCs while education on referrals procedures of patients with cancer and COVID-19 were more applied at the RH.

The latter could be because hospitals already have protocols for other types of infectious diseases which could give an advantage in COVID-19 patient management and a need to best instruct PHCs staff in this matter. Work schedules in PHCs were extended due to the increased number of patients seeking medical attention during the pandemic as hospitals were already working “around the clock” before the pandemic. Regarding cancer and COVID-19 patient referrals, RH staff had a need to re-adapt the approach to ensure timeliness and patient safety, hence the need to better understand how to do so since they directly attend this population, compared to PHC staff.

This re-organization occurred because, in our local health unit, there was no special taskforce to treat COVID-19 cancer patients; the same staff involved with oncology prior the pandemic continued with the OPM services. Moreover, some studies point that the humoral response to COVID-19 vaccination in cancer patients can be influenced by the type of malignancy (especially hematological which undergo somewhat more significant immunosuppression), the type of anti-cancer treatment and demographic factors like age and gender. We consider that an infectiologist should have been integrated in the oncology multidisciplinary team to better support their immunosuppressed cancer patients [36,37,38,39].

Other items that were not statistically significant, nonetheless, indicating some level of difference between groups response rates were symptom monitorization in patients with active cancer treatment and segregation of early detection areas. Regarding the first item, RH staff had higher response rates since it is at hospitals where they receive their treatment, and their condition is mostly managed. Early detection areas had higher response rates in PHCs, indicating a greater effort to detect and manage infected patients in healthcare platforms near patients’ residences.

The Cronbach’s alpha analysis for internal consistency shows one acceptable value of 0.723 for Table 1 and three good values of 0.833, 0.855 and 0.828 for Table 2, Table 3 and Table 4, respectively. The latter shows reliability in the way the checklists were constructed, allowing us to measure the same underlying dimension of adjustments of OPM during the COVID-19 pandemic.

As mentioned before, our analysis of recommendations for the RH pharmacist checklist could not undergo the same analysis as the others. Nonetheless, Appendix A shows the answers and response rates from the responders. Their answers confirm a contingency plan that did not reflect an impact on the pharmaceutical services logistics, since most responders did not apply the proposed adjustments. The recommendations with higher response rates were an extension of drug dispensing periods, drug acquisition processes, communication with healthcare centers when access to the hospital is difficult and increased drug stocks in the cancer day hospital.

COVID-19 will not be the last pandemic and this paper could shed light on the various aspects of malignant disease management during future pandemics. These aspects could help build proper guidelines in accordance with the healthcare systems. The analysis of our results shows, in general terms, that the healthcare professionals involved with cancer patients in our local unit were aware of the published guidelines and recommendations, as well as their application in their practice.

## 5. Conclusions

With the use of our checklists and a Mann–Whitney analysis, we were able to determine the level of agreement and cohesiveness nurses and physicians at a local health unit had in regards the management of their oncology patients during the COVID-19 pandemic and the recommendations given by groups of experts.

The checklists helped determine what was considered essential and/or available for the proper functioning of the OPM at this local health unit. Therefore, hospital responders agreed on containing and preventing all or most possible ways of infection through physical changes that limit contact and time of contact with patients followed by oncology patient management measures such as adjustments in treatments, analysis, patient care, referrals and emergencies. The same conclusion can be withdrawn from PHC responders followed by OPM changes in work schedules, teamwork, patient care and patient referral. The latter are to ensure that every other service, including oncology, continues to provide their services to those in need.

As a local health unit, there is an agreement regarding most of the applied OPM measures since only three recommendations showed statistically significantly different *p* values which relate to different forms of work and flow of patients between institutions. Regarding pharmaceutical services involved with the oncology service, the agreement between responders implied the continuation of work as before the pandemic.

In conclusion, the COVID-19 pandemic demanded changes in patient management, especially for the vulnerable such as cancer patients. Our checklists show how healthcare platforms and professionals manage a specific patient population throughout a world pandemic with acceptable-to-good levels of internal consistency and reliability. Groups analysis allow a proper view of what matters most to health professionals to provide their services. Our local health unit, as an RH, and the PHCs seem to agree on which measures were important, realistic, and available for the sake of their oncology patients.

### Study Limitations

During the pandemic there was low professional availability to first meet and receive a first introduction on the work objectives, resulting in longer work periods. In addition, this paper collects and analyzes only three classes of healthcare professional appreciation regarding OPM which limits the overview of the situation as there are more professionals involved (such as nutrition, physical therapy, psychology). We think that the perception of the patients’ own process of disease management should be studied as well.

## Figures and Tables

**Table 1 ijerph-19-09041-t001:** Physical Changes in the RH regarding OPM During the COVID-19 Pandemic.

Recommendation	Healthcare Professional (N = 33)	*N*	Mean Rank	Response Rate for Developed Recommendation (%)	Mann–Whitney U-Test	*p* Value
1. Decrease in the number of routine visits	Nurse	18	15.58	63.6	109,500	0.361
	Physician	15	18.70			
2. Adequate access to service resources (material and human)	Nurse	18	14.08	54.5	82,500	0.057
	Physician	15	20.50			
3. Implementation of remote consult systems	Nurse	18	17.75	54.5	148,500	0.630
	Physician	15	16.10			
4. Service entry and exit point	Nurse	18	16.33	51.5	123,000	0.682
	Physician	15	17.80			
5. Temperature measurement at Day Hospital entrance	Nurse	18	14.50	51.5	90,000	0.108
	Physician	15	20.00			
6. Telemedicine consultations	Nurse	18	18.00	39.4	153,000	0.532
	Physician	15	15.80			
7. Work in “mirror teams”	Nurse	18	13.75	30.3	76,500	0.033
	Physician	15	20.90			
8. Analysis and treatments schedules	Nurse	18	16.00	27.3	117,000	0.532
	Physician	15	18.20			
9. Care education for patients with immunosuppressive treatment	Nurse	18	16.92	27.3	133,500	0.957
	Physician	15	17.10			
10. Reorganization of the diagnostic and therapeutic referrals at the hospital	Nurse	18	17.33	24.2	141,000	0.845
	Physician	15	16.60			
11. Provide laboratory analyses in locations closer to the patient’s residence	Nurse	18	16.83	21.2	132,000	0.929
	Physician	15	17.20			
12. Incentive to multidisciplinary videoconference consultations	Nurse	18	17.25	18.2	139,500	0.873
	Physician	15	16.70			
13. Extension of work schedules	Nurse	18	15.83	15.2	114,000	0.464
	Physician	15	18.40			
14. Segregation of early detection areas	Nurse	18	17.17	12.1	138,000	0.929
	Physician	15	16.80			
15. Local and schedule of blood and biological product samples	Nurse	18	15.75	9.1	112,500	0.421
	Physician	15	18.50			
16. Design of new clinical pathways for cancer patients	Nurse	18	17.08	6.1	136,500	0.957
	Physician	15	16.90			

RH: Regional Hospital, OPM: Oncology Patient Management.

**Table 2 ijerph-19-09041-t002:** Technical–Scientific Changes at the RH regarding OPM During the COVID-19 Pandemic.

Recommendation	Healthcare Professional (N = 33)	*N*	Mean Rank	Response Rate for Developed Recommendation (%)	Mann-Whitney U-Test	*p* Value
1. Strict measures of infection control and prevention	Nurse	18	16.25	78.8	121,500	0.630
	Physician	15	17.90			
2. Follow-up (video consultations or phone calls. analysis at home or area of residence)	Nurse	18	17.08	72.7	136,500	0.957
	Physician	15	16.90			
3. Personal protective equipment (PPE) provided and training on its use	Nurse	18	17.08	72.7	136,500	0.957
	Physician	15	16.90			
4. Implementing triage protocols for COVID-19 symptoms	Nurse	18	17.00	66.7	135,000	1.000
	Physician	15	17.00			
5. Delay visits and surveillance exams in patients considered non-urgent	Nurse	18	20.58	60.6	199,500	0.018
	Physician	15	12.70			
6. Symptomatology assessment considering other etiologies in cancer	Nurse	18	15.50	57.6	108,000	0.343
	Physician	15	18.80			
7. Education on suspected cases identification. isolation procedures and sample collection	Nurse	18	15.42	51.6	106,500	0.307
	Physician	15	18.90			
8. Adapted and communicated risk assessment to your healthcare team	Nurse	18	15.33	45.5	105,000	0.290
	Physician	15	19.00			
9. Test patients undergoing imaging analysis and/or requiring urgent surgery	Nurse	18	19.00	45.5	171,000	0.202
	Physician	15	14.60			
10. Education on cancer patients and COVID-19 positive referral procedures	Nurse	18	15.75	42.4	112,500	0.421
	Physician	15	18.50			
11. Monitorization of symptoms in patients with active cancer treatment	Nurse	18	17.08	39.4	136,500	0.957
	Physician	15	16.90			
12. Oncology Patient Management according to risk and treatment	Nurse	18	17.92	33.3	151,500	0.556
	Physician	15	15.90			
13. Attending oncological emergencies as long as they do not affect the patient vital prognosis	Nurse	18	16.92	27.3	133,500	0.957
	Physician	15	17.10			
14. Stricter surveillance for high-risk cases	Nurse	18	15.92	21.2	115,500	0.486
	Physician	15	18.30			
15. Favor oral medication treatment schemes	Nurse	18	16.83	21.2	132,000	0.929
	Physician	15	17.20			
16. Prescription of oral treatments for longer periods (2–3 months)	Nurse	18	16.33	18.2	123,000	0.682
	Physician	15	17.80			
17. Spacing visits for patients with hormone therapy and stable disease	Nurse	18	15.42	18.2	106,500	0.307
	Physician	15	18.90			
18. Priority access to surgery for eligible patients	Nurse	18	17.67	15.2	147,000	0.682
	Physician	15	16.20			
19. Implementing stricter criteria for complementary treatments to chemotherapy in low-risk patients	Nurse	18	16.75	15.2	130,500	0.873
	Physician	15	17.30			
20. Pause of treatments and/or surveillance in stable patients	Nurse	18	17.67	15.2	147,000	0.682
	Physician	15	16.20			
21. Medication schemes change to oral drugs	Nurse	18	17.17	12.1	138,000	0.929
	Physician	15	16.80			
22. Use of treatments with less hematological toxicity and less immunosuppression	Nurse	18	17.17	12.1	138,000	0.929
	Physician	15	16.80			
23. Hormone therapy spacing	Nurse	18	15.33	12.1	105,000	0.290
	Physician	15	19.00			
24. Increase indications for growth factors as supportive therapy	Nurse	18	15.75	9.1	112,500	0.421
	Physician	15	18.50			
25. Cytotoxic treatment switch with alternative drugs	Nurse	18	16.17	6.1	120,000	0.605
	Physician	15	18.00			
26. Provide at home use of chronic hormone therapy	Nurse	18	16.17	6.1	120,000	0.605
	Physician	15	18.00			
27. Avoid “dose-dense” schemes	Nurse	18	17.50	3.0	144,000	0.762
	Physician	15	16.40			
28. Ensure patients access to clinical trials	Nurse	18	17.00	0	136,000	1.000
	Physician	15	17.00			

RH: Regional Hospital, OPM: Oncology Patient Management.

**Table 3 ijerph-19-09041-t003:** Changes in the OPM Processes During the COVID-19 Pandemic (PHCs).

Recommendation	Healthcare Professional (N = 75)	*N*	Mean Rank	Response Rate for Developed Recommendation (%)	Mann- Whitney U-Test	*p* Value
1. PPE provided and training on its use.	Nurse	46	38.71	86.7	699,500	0.548
	Physician	29	36.88			
2. Strict measures of infection control and prevention	Nurse	46	37.15	76.0	628,000	0.566
	Physician	29	39.34			
3. Implementing triage protocols for COVID-19 symptoms	Nurse	46	40.73	74.7	792,500	0.070
	Physician	29	33.67			
4. Education on suspected cases identification. isolation procedures and sample collection	Nurse	46	37.78	73.3	657,000	0.887
	Physician	29	38.34			
5. Decrease in the number of routine visits	Nurse	46	42.12	65.3	856,500	0.012
	Physician	29	31.47			
6. Delay of visits and surveillance exams in patients considered non-urgent	Nurse	46	40.70	50.7	791,000	0.119
	Physician	29	33.72			
7. Adapted and communicated risk assessment to your healthcare team	Nurse	46	38.57	49.3	693,000	0.744
	Physician	29	37.10			
8. Implementation of remote consult systems	Nurse	46	41.33	48.0	820,000	0.054
	Physician	29	32.72			
9. Oncology Patient Management according to risk and treatment	Nurse	46	38.70	45.3	699,000	0.686
	Physician	29	36.90			
10. Adequate access to service resources (material and human)	Nurse	46	36.88	42.7	615,500	0.513
	Physician	29	39.78			
11. Extension of work schedules	Nurse	46	36.01	36.0	575,500	0.231
	Physician	29	41.16			
12. Work in “mirror teams”	Nurse	46	38.53	25.3	691,500	0.723
	Physician	29	37.16			
13. Segregation of early detection areas	Nurse	46	38.03	24.0	668,500	0.982
	Physician	29	37.95			
14. Monitorization of symptoms in patients with active cancer treatment	Nurse	46	38.85	24.0	706,000	0.566
	Physician	29	36.66			
15. Incentive to multidisciplinary videoconference consultations	Nurse	46	36.72	22.7	608,000	0.376
	Physician	29	40.03			
16. Education on cancer patients and COVID-19 positive referral procedures	Nurse	46	37.85	21.3	660,000	0.915
	Physician	29	38.24			
17. Design of new clinical pathways for cancer patients	Nurse	46	38.92	13.3	709,500	0.432
	Physician	29	36.53			

PHCs: Primary Healthcare Centers, OPM: Oncology Patient Management.

**Table 4 ijerph-19-09041-t004:** Changes in the OPM Processes During the COVID-19 Pandemic (Local Health Unit).

Recommendation	Healthcare Platform (N = 108)	*N*	Mean Rank	Response Rate for Developed Recommendation (%)	Mann-Whitney U-Test	*p* Value
PPE provided and training on its use.	PHCs	75	52.20	82.4	1410,000	0.081
	RH	33	59.73			
2. Strict measures of infection control and prevention	PHCs	75	54.96	76.9	1203,000	0.753
	RH	33	53.45			
3. Implementing triage protocols for COVID-19 symptoms	PHCs	75	53.18	72.2	1336,500	0.395
	RH	33	57.50			
4. Education on suspected cases identification. isolation procedures and sample collection	PHCs	75	50.90	66.7	1507,500	0.027
	RH	33	62.68			
5. Decrease in the number of routine visits	PHCs	75	54.22	64.8	1258,500	0.866
	RH	33	55.14			
6. Delay of visits and surveillance exams in patients considered non-urgent	PHCs	75	56.14	53.7	1114,500	0.342
	RH	33	50.77			
7. Implementation of remote consult systems	PHCs	75	55.58	50.0	1156,500	0.533
	RH	33	52.05			
8. Adapted and communicated risk assessment to your healthcare team	PHCs	75	53.86	48.1	1285,000	0.711
	RH	33	55.95			
9. Adequate access to service resources (material and human)	PHCs	75	56.46	46.3	1090,500	0.256
	RH	33	50.05			
10. Oncology Patient Management according to risk and treatment	PHCs	75	52.52	41.7	1386,000	0.246
	RH	33	59.00			
11. Extension of work schedules	PHCs	75	51.06	29.6	1495,500	0.030
	RH	33	62.32			
12. Monitorization of symptoms in patients with active cancer treatment	PHCs	75	57.04	28.7	1047,000	0.105
	RH	33	48.73			
13. Education on cancer patients and COVID-19 positive referral procedures	PHCs	75	57.98	27.8	976,500	0.025
	RH	33	46.59			
14. Work in “mirror teams”	PHCs	75	55.32	26.9	1176,000	0.593
	RH	33	52.64			
15. Incentive to multidisciplinary videoconference consultations	PHCs	75	53.76	21.3	1293,000	0.602
	RH	33	56.18			
16. Segregation of early detection areas	PHCs	75	52.54	20.4	1384,500	0.160
	RH	33	58.95			
17. Design of new clinical pathways for cancer patients	PHCs	75	53.30	11.1	1327,500	0.270
	RH	33	57.23			

Primary Healthcare Centers: PHCs, Regional Hospital: RH.

## Data Availability

The data presented in this study are available upon request from the corresponding author.

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
