# Peer review of "The Effects of the COVID-19 Pandemic in Oncology Patient Management"

_ijerph, 2022, doi:10.3390/ijerph19159041_

Round 1

Reviewer 1 Report

An interesting article exploring a timely topic in oncology in these times.

Some changes are necessary:

1. A linguistic revision is necessary

2. The discussion should be expanded, and the limitations of the current paper described

3. The authors should expand the introduction section, adding some recently published papers regarding covid19 and cancer (PMID: 35109688PMID: 32658591 )

Major changes are suggested

Author Response

Dear Reviewer,

Thank you very much for all the suggestions for our paper. All the authors gathered to discuss and respond thoroughly to all the comments.

  1. A linguistic revision is necessary: The authors revised and checked the linguistics of all the manuscript for a better understanding of the concepts, contexts and processes involved in the manuscript. Also, we double checked with one native-speaking colleague for better insight on the paper understanding.
  2. The Discussion should be expanded, and the limitations of the current paper described: In line 363 we used some of the suggested articles to expand on some topic considered on those references. We added one more paragraph using those references and added some closing thoughts on the discussion (line 396). Regarding the limitations of the paper, they are described, as suggested, in lines 436-441.
  3. The authors should expand the introduction section, adding some recently published papers regarding COVID-19 and cancer: We expand the introduction section using the suggested references given by the reviewer. From line 36 until line 60 of the introduction. 

I hope we answered all the suggested revisions for your consideration regarding the publication of this paper. Please let us know if any further corrections are needed. Thank you for your consideration and attention.

Best Regards,

Dr. Mario Forrester Quesada

PhD Student Universidade da Beira Interior, Covilhã, Portugal

Reviewer 2 Report

Dear authors, 

Thank you for submitting the paper entitled "The effects of the COVID-19 Pandemic in Oncology Patient Management". From the beginning COVID-19 pandemic several risks have been imposed either on cancer patients or the centers providing the healthcare facilities.  After two years, the pandemic has been controlled in most areas of the world and new emergencies (such as Russia vs Ukraine War) have replaced it. Therefore, it can not be considered as a novel topic. However, COVID-19 will not be the last pandemic and this paper could shed light on the various aspects of malignant disease management during the other pandemic in the future. 

Here you may find my comment. Please consider them to improve the quality of your paper. Please be informed that citing the suggested referenced although is appreciated, is not mandatory. And I only provide them to you to have an idea about what is important in different parts of your paper. Therefore, these papers have been suggested to improve the manuscript quality in accordance with the most up-to-date literature and there was no intention to induce citation and you can use them optionally based on your preference.

--------------------------------------------------------------------------

Introduction and Discussion 

Please consider the results of these papers in the introduction. These papers have shown  that COVID-19 related mortality in patients with underlying disease such as cancer patients is considerable though the symptoms may not be as they are expected. Moreover, the third paper showed that almost 20% of these patients may suffer from an asymptomatic disease  was evident only in serological assessment. The last one shows the impact of COVID-19 pandemic on the oncologic centers.   

1. COVID ‐19 in cancer patients may be presented by atypical symptoms and higher mortality rate, a case‐controlled study from Iran http://dx.doi.org/10.1002/cnr2.1378

2. COVID-19 in Cancer and non-Cancer Patients http://dx.doi.org/10.5812/ijcm.110907

  3. Multicenter Study of Antibody Seroprevalence against COVID-19 in Patients Presenting to Iranian Cancer Centers after One Year of the COVID-19 Pandemic http://dx.doi.org/10.1080/07357907.2021.1995742   4. The Covid-19 Outbreak And Oncology Centers In Iran https://dx.doi.org/10.5812/ijcm.103283
Moreover, these papers can be used in the discussion to discuss the humoral responses after vaccination in patients with underlying disease such as cancer patients:
1. The first manuscript in world on the Immunogenicity and Safety of the Inactivated SARS-CoV-2 Vaccine (BBIBP-CorV) in Patients with Malignancy http://dx.doi.org/10.1080/07357907.2021.1992420

2. A Cohort Study on the Immunogenicity and Safety of the Inactivated SARS-CoV-2 Vaccine (BBIBP-CorV) in Patients With Breast Cancer; Does Trastuzumab Interfere With the Outcome? https://doi.org/10.3389/fendo.2022.798975
3. COVID-19 Vaccination in Patients with Malignancy; A Systematic Review and meta-analysis of the Efficacy and Safety https://doi.org/10.3389/fendo.2022.860238
The two first studies showed that immune response (seroconversion) rates following COVID-19 vaccination in patients with underlying disease such as cancer patients are compromised in older patients, those with hematological malignancies and chemotherapy receivers.  -------------------------------------------------------------------------- Materials and Methods: Please write the type of study. Please report the status of covid-19 in the are where the study had been conducted, especially regarding the  changes in the number of health care staff, their leaves, their shifts and so on. Please describe the validity and reliability of checklists.  Please write the inclusion and exclusion criteria clearly.    -------------------------------------------------------------------------- Results: Please report the demographic data as well (gender, age, work experience [year], working shifts/hours per day, and so on) Please compare your findings based on the demographic data as well

Author Response

Dear Reviewer,

Thank you for your consideration and attention towards my manuscript and the revisions given. All the authors gathered to discuss and respond thoroughly to all the comments, as follows:

  1. Introduction: we incorporated the first 3 of the 4 papers suggested in order to explain and expand on the introductory ideas of COVID-19 mortality and patients with cancer (Lines 62-72).
  2. Materials and Methods: as suggested we specified the type of study in line 86. The report status of covid-19 in the area of study, the changes in the number of health care staff, their leaves, their shifts and so on were reported (Lines 96-113), however, the information available was of the Portuguese National Health System reports and specifical information from our local health unit was not available upon demand to the administrative services. The validity and reliability of checklists was expanded and explained as suggested by the reviewer in lines 126-137 and lines 160-164. Additionally, the inclusion and exclusion criteria were described in lines 166-175.
  3. Results: regarding the requested demographic data of our responders and the comparison with our findings, we were not able to respond to this suggested revision since the answers of the key deciders were anonymous in order to provide a safe and private environment to our responders in other sections of the questionnaire that required an open critic space. All the latter is explained in the Patient and Public Involvement section (lines 178-188)
  4. Discussion: in this section we added content to the discussion using all the references suggested by the author to discuss the humoral response after vaccination in patients with underlying disease such as cancer patients which are compromised in older patients, those with hematological malignancies and chemotherapy receivers (Lines 363-371)

I hope we answered all the suggested revisions for your consideration regarding the publication of this paper. Please let us know if any further corrections are needed. Thank you for your consideration and attention.

Best Regards,

Dr. Mario Forrester Quesada

PhD Student Universidade da Beira Interior, Covilhã, Portugal

Round 2

Reviewer 1 Report

Acceptance.

Reviewer 2 Report

Thank you for all the modifications which you have provided.